# Association between cannabis use and result of COVID-19 testing among scholarised adolescents in Andalusia: A cross-sectional study

**María-Carmen Torrejón-Guirado**[1], **Miguel Ángel Baena-Jiménez**[2*], **Marta Lima-Serrano**[1]

**1** Department of Nursing, School of Nursing, Physiotherapy, and Podiatry, University of Seville, Institute of Biomedicine of Seville (IBiS), Seville, Spain, **2** Department of General and Specialist Surgery, Hospital Universitario Virgen Macarena, Spain

* mang.baena.sspa@juntadeandalucia.es

## Abstract

### Aims

The COVID-19 pandemic may have impacted cannabis use among the adolescent population. Therefore, it is important to understand the potential factors linking cannabis use to COVID-19 and vice versa, as well as the consumption patterns following the onset of COVID-19.

### Methods

This study conducted in Andalusia, Spain. Although, 1,051 adolescents aged 14–18 years were included in this cross-sectional study, of these, the 89 (8.5%) reported testing positive for COVID-19, who were the main target for our analyses. Sociodemographic, psychological, and cannabis use variables, as well as COVID-19 testing positive, isolation and some perceptions about cannabis and COVID-19 were analysed. A binary logistic regression, was conducted to examine the association between COVID-19-related factors and controlled for sociodemographic and psychological variables, assessed estimations with cannabis use.

### Results

Adolescents testing positive for COVID-19 are almost 2.89 times more likely to have used cannabis in the last month. Additionally, number of isolations was higher among cannabis users and positive testing. Cannabis users showed a lack of perceived risk that cannabis use could exacerbate COVID-19 and that cannabis' smoke could spread COVID-19. Sharing cannabis among adolescents during the pandemic may increase the risk of COVID-19 transmission.

**Data availability statement:** The data underlying the results presented in the study are available from Zenodo (a public repository; accession DOI: https://doi.org/10.5281/zenodo.15106665).

**Funding:** This work was supported by the University of Seville research program (VPPI-US) in terms of a pre-doctoral contract of Torrejón-Guirado. The funder had no role in study design, data collection and analysis, decision to publish, or preparation of the manuscript. Additionally, this was a secondary objective of a nationally funded project by the Spanish Ministry of Science, Innovation, and Universities under the "State Plan 2017-2020 Challenges - R&D&I Projects" call, titled "Cannabis Alert: Evaluation of a Tailored Computer-Based Intervention for the Prevention of Cannabis Use in Adolescents Aged 14 to 18 Years" (PID2019-107229RA-I00). This funder also had no role in study design, data collection and analysis, decision to publish, or preparation of the manuscript.

**Competing interests:** The authors have declared that no competing interests exist.

## Conclusions

Informing whether adolescents who use cannabis may be more likely to test positive for COVID-19 can contribute to a better understanding of substance use patterns during a public health emergency. However, given the exploratory and cross-sectional nature of this study, as well as its limited and context-specific sample, the findings should be interpreted with caution. Despite this, we believe that this research may inform future studies exploring the impact of cannabis use among adolescents during public emergency situations in larger samples and other contexts.

## Introduction

In March 2020, Spain declared a nationwide lockdown to contain the rapid spread of COVID-19. These measures led to profound healthcare and social disruptions, significantly affecting adolescents' daily lives, including mobility restrictions, school closures, and reduced social interaction [1].

The World Health Organization (WHO) declared the end of the pandemic at the beginning of May 2023 [2]. However, during the pandemic, reported cases of COVID-19 were substantial [3]. Certain population groups, such as adolescents, tend to experience this disease in an asymptomatic manner or with milder symptoms [4,5], which may have complicated detection and reporting of positive cases in this population.

Moreover, the United Nations' Annual Report for 2020 against drug abuse highlighted that most countries reported an increase in cannabis consumption during the COVID-19 pandemic [6]. Previous studies have reported that cannabis use was associated with an increased risk of hospitalization and ICU admission among individuals with COVID-19 [7]. Nevertheless, the extent to which cannabis use is associated with reporting testing positive for COVID-19 among adolescents remains insufficiently examined, particularly in Spain.

Cannabis consumption stands as the most widely used illicit drug among teenagers globally [8]. The global prevalence of cannabis use in the last year was 5.8% (corresponding to 12 million) among 15- and 16-year-old adolescents [9].

In Spain, adolescents aged 14–17 years reported that the prevalence of cannabis use is 20.7% and increasing to 35.6% at age 18. In Andalusia specifically, according to ESTUDES 2025 survey, the average age of initiation of cannabis use is 15.1 years. Furthermore, lifetime prevalence is 21%, and 11.6% report use in the past 30 days. Approximately 94% of users consume cannabis in the form of hand-rolled cigarettes. In Spain, these are typically prepared by mixing cannabis (frequently in the form of hashish) with tobacco [10].

However, it is worth noting that in previous reports from 2022, which analyzed the impact of the pandemic in Andalusia, ESTUDES survey reported a decline in cannabis use, attributing this primarily to the pandemic (e.g., due to mobility restrictions) [10]. These data coincide with the OEDA-COVID 2020 survey, which was conducted on the general population residing in Spain, with fieldwork carried out between

November 10 and December 3, 2020 [11]. This telephone survey identified a change in cannabis consumption patterns, revealing that 3.5% of the population either stopped using cannabis or reduced their consumption during the pandemic, with this percentage being higher among men than women. However, ESTUDES 2022 reported that cannabis use prevalence had progressively increased as social outings, particularly nighttime activities, had resumed [12].

Despite this evolving evidence, few studies have specifically examined the association between cannabis use and reporting testing positive for COVID-19 among adolescents. A recent systematic review summarize cannabis use among adolescents during the COVID-19 pandemic [13]. Yet, few studies give importance in the first instance to the COVID-19 testing positive. For instance, other scoping review explored the impact of cannabis consumption on COVID-19 in a predominantly adolescent population, concluding that the validity of studies conducted thus far was insufficient to establish accurate associations [14]. International studies emphasized the importance of investigating the association between COVID-19 testing positive and cannabis consumption in adolescents [15,16]. However, no studies were found that linked COVID-19 and recreational cannabis use in adolescents in Spain.

Given that COVID-19 is still prevalent, and cannabis consumption continues to rise each year, it is necessary to conduct studies in this context to explore factors related to COVID-19 and cannabis consumption in adolescents.

## The study

To address this gap, a cross-sectional study was conducted in Sevilla, Cádiz, Córdoba, and Huelva. Andalusia is the largest autonomous community in Spain, covering approximately 87,600 km², with a diverse population spread across urban and rural areas [17]. This geographical and demographic variability poses challenges for conducting studies, including participant recruitment and accessibility, so that we decided to include Western Andalusian participants.

The hypothesis generation posits that the result of COVID-19 testing positive among Western Andalusian adolescents will have a positive epidemiological correlation with cannabis use. Yet, the variables that are related to this hypothesis may differ per frequency of cannabis use, age, and other factors. Therefore, the objectives of this study are: (1) to identify putative cannabis use associated with COVID-19 testing positive in a sample of Andalusian students aged 14–18 and (2) to explore correlations between cannabis use and testing positive COVID-19 and behavioural variables related with COVID-19 (such as isolations and sharing cannabis during the pandemic) as well as perceptions about the relationship of COVID-19 and cannabis use.

## Materials and methods

### Design and population

A random sample was collected from the clusters of 21 high schools from both rural and urban settings of four provinces of Western Andalusia. The sample size was calculated using the online tool GRANMO (Result 1046, accepting an Alpha risk of 0.05 and Beta of 0.2 in a bilateral test). The sample was randomly selected through cluster sampling. The recruitment started on 01-05-2021 and the end of the recruitment period for this study was 30-06-2021.

Inclusion criteria were adolescents aged 14–18 years, and adolescents from educational level of 9th–12th grade or vocational training (VT). Duplicate or incomplete questionnaires were excluded. Responses were obtained from a total of 1086 students, of which 1051 met the inclusion criteria (484 male students and 567 female students) (Fig 1).

### Measurements

For better comprehension, the variables were divided into three groups: the sociodemographic and psychological variables, behavioural variables (such as isolations or sharing cannabis during the COVID-19), and the more subjective variables related to COVID-19, referring to the adolescent#39;s perceptions. In addition to the two main outcomes of the article: testing COVID-19 and cannabis use.

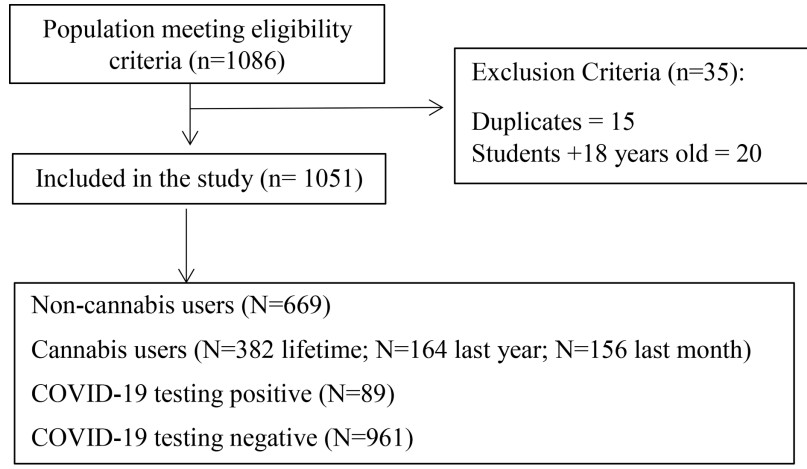

**Fig 1. Recruitment Process Flowchart Based on the Strengthening the Reporting of Observational Studies in Epidemiology (STROBE) Statement [18].**

**Testing COVID-19.** The following ad hoc questions were designed to explore the relationship of COVID-19 to cannabis use: COVID-19 testing positive (0 = no, 1 = yes).

**Cannabis consumption.** The Spanish ESTUDES survey was used to assess cannabis consumption by asking two different questions: "Do you consume cannabis?," and "Have you ever consumed cannabis/marijuana, even if it was just a puff from a joint among friends ["In your life," "In the last 12 months," and "In the last 30 days"]?" Responses were coded as follows: (0 = no cannabis use), (1 = I used cannabis 1 day), (2 = 2 days), (3 = 3 days), (4 = 4–5 days), (5 = 6–9 days), (6 = 10–19 days), and (7 = 20 days or more) [19].

**Sociodemographic and psychological variables.** Age (in years), gender (female (1) or male (0)), educational level (9th–12th grade or VT) and academic performance (0%–39%, 50%–59%, 60%–69%, 70%–79%, 80%–100%).

Rosenberg Self-Esteem Scale with 10 questions (1 "strongly disagree" to 4 "strongly agree"; reverse coding: items 2, 6, 8, and 9) (Cronbach's $\alpha = 0.459$) [20], and *personality*, assessed using the Substance Use Risk Profile Scale with 23 questions (1 "strongly disagree" to 4 "strongly agree"; reverse coding: items 1, 4, 7, 13, 20, and 23) and with a four-dimensional component: hopelessness, impulsiveness, sensation seeking, and sensitivity anxiety (Cronbach's $\alpha = 0.737$) [21].

**Behavioural variables.** Number of isolation (0–10). Have you shared cannabis during the COVID-19 pandemic? (0 = no, 1 = yes).

**Perception variables associating COVID-19 with cannabis use.** Do you think that cannabis worsen COVID-19? (0 = no, 1 = yes); Do you think that cannabis smoke spread COVID-19? (0 = no, 1 = yes). Perception variables related to the consumption of other substances from the OEDA-COVID 2020 survey served as a reference for designing our questions [22,23].

## Procedure

Using the Andalusian Regional Government website, a comprehensive list of all schools in the mentioned provinces was obtained. Randomly, an equal number of public and private high schools, as well as an equal number of rural and urban secondary education institutions (IES), were selected. Following this selection, telephonic and email contact was established with the directors of 87 IES to request their collaboration, with responses received from 20 directors. Subsequently, based on the school#39;s protocols, directors directed us to the guidance department, facilitating our communication with the guidance counsellors. Preliminary information regarding this study, along with the informed consent and questionnaire,

was dispatched to the school directors and guidance counsellors. Once their participation was confirmed, specific dates were arranged to conduct the questionnaires, allowing us to collect data from the selected sample. This process occurred via virtual platform (Le Sphinx) or in-person visits, consistently during school hours and with oversight from responsible teachers to ensure the agreed-upon procedures were carried out with the utmost assurance.

### Participants consent and ethics statement

All participants provided their written informed consent. Participants were instructed that they could chose not to complete the survey. Active written consent was requested from parents, unless the high school indicated that they wanted to provide passive written consent. Parents were notified in writing at least one week in advance of data collection, so they could opt out their adolescent from data collection.

This observational study was approved by the Andalusian Research Ethics Committee (registration number: 0073-N-18) in 2018. The confidentiality of the data was guaranteed and explained to the participants and their parents, and the procedures followed the Regulation (EU) 2016/679 of the European Parliament and the Data Protection Council. In addition, during the design of the study, the data protection delegate of the University of Seville was contacted to carry out the corresponding procedures on the treatment of data of minors. This study was performed in line with the principles of the Declaration of Helsinki, and its later amendments or comparable ethical standards.

### Statistical analyses

To verify the normal distribution of the selected variables, the Kolmogorov-Smirnov test was conducted. A descriptive study was performed to calculate frequencies and percentages for qualitative variables, and means and standard deviations for quantitative variables. Additionally, a bivariate analysis was conducted: the Chi-Square test was used to determine the independence between two qualitative variables, and the T-Student test was employed to compare the independence between a quantitative and a qualitative variable.

To assess how well the measured factors are associated with cannabis use, binary logistic regression analysis was conducted [24]. All variables were included in the model, except for the educational level, because it had the largest p-value in the chi-square test. Personality and self-esteem were used as index because of their multidimensional nature. The level of significance was set at $p < 0.05$, and the enter method was used in this study. As there were very few missing values, no imputation data was performed. This statistical analysis was conducted using Statistical Package for the Social Sciences [25].

## Results

### Study sample characteristics and COVID-19 testing positive

Out of a total of 1,051 adolescent participants with an average age of 15.29 years, where 27% (N = 284) of the participants were 14 years old, 29.9% (N = 314) were 15 years old, 24.9% (N = 262) were 16 years old, 15.3% (N = 161) were 17 years old, and 2.9% (N = 30) were 18 years old. Moreover, 53.9% (N = 567) were female.

The academic performance of the participants showed an average score of 2.10, that indicates that the majority of participants demonstrated good academic performance. Participants reported having an average self-esteem score of 28.78 (SD = 3.936), indicating a moderate level of self-esteem among the surveyed students, with no severe self-esteem issues observed (see Table 1).

Regarding the Risk Profile for Substance Use, the average scores were Hopelessness (M = 21.04, SD = 3.444), Impulsivity (M = 11.86, SD = 3.207), Anxiety (M = 12.091, SD = 3.226), and Sensation Seeking (M = 16.63, SD = 3.968). There was no specified cut-off points provided by the author, nor any other studies to use as a reference, and due to the absence of a comparison group, it is not possible to determine if these values are low, medium, or high.

**Table 1. Sociodemographic, psychological, and related COVID-19 and cannabis behaviour and perception description of the sample and its correlation with adolescents' COVID-19 testing positive.**

| Variables | Participants N = 1051 | Negative COVID-19 N = 961 | Positive COVID-19 N = 89 | T-test o χ2 | P valor of t-test o χ2 |
|---|---|---|---|---|---|
| *Sociodemographic and psychological* | | | | | |
| **Schools** | N = 20 | | | | |
| **Age** (14–18) (mean, SD) (m 9) | 15.29 (1.289) | 15.29 (1.302) | 15.22 (1.149) | 0.520 | 0.603 |
| **Gender** (m 0) | | | | | |
| • Male | 484 (46.1%) | 440 (45.8%) | 44 (49.4%) | 0.437 | 0.291 |
| • Female | 567 (53.9%) | 521 (54.2%) | 45 (50.6%) | | |
| **Adolescents Educational Level** (m 0) | | | | | |
| • 10th Grade | 697 (66.3%) | 635 (66.1%) | 62 (69.7%) | 0.469 | 0.791 |
| • 11th Grade | 302 (28.7%) | 278 (28.9%) | 23 (25.8%) | | |
| • Vocational Training (VT) | 52 (5%) | 48 (5.0%) | 4 (4.5%) | | |
| **Academic Performance** (mean, SD) (m 1) | 2.10 (0.970) | 2.09 (0.971) | 2.31 (0.937) | −2.127 | **0.034** |
| **Self-esteem** (mean, SD) (m1) | 28.78 (3.936) | 28.76 (3.929) | 29.00 (4.009) | −0.551 | 0.582 |
| **Risk Profile of Substance Use** (mean, SD) (m 1) | | | | | |
| • Hopeleness | 21.04 (3.444) | 20.99 (3.439) | 21.69 (3.459) | −1.833 | 0.067 |
| • Impulsivity | 11.86 (3.207) | 11.90 (3.192) | 11.44 (3.364) | 1.288 | 0.198 |
| • Sensation Seeking | 16.63 (3.968) | 16.58 (3.964) | 17.21 (4.001) | −1.441 | 0.150 |
| • Axiety | 12.09 (3.226) | 12.12 (3.174) | 11.69 (3.740) | 1.044 | 0.299 |
| *Behavioural* | | | | | |
| **Isolation** (mean, SD) (m 1) | 0.74 (0.968) | 0.68 (0.949) | 1.36 (0.956) | −6.444 | **<0.001** |
| **Sharing Cannabis in Pandemic** (m 158) | | | | | |
| • No | 893 (93.0%) | 818 (93.1%) | 75 (92.6%) | 0.025 | 0.506 |
| • Yes | 67 (7.0%) | 61 (6.9%) | 6 (7.4%) | | |
| *Perceptions* | | | | | |
| **Cannabis' Smoke Contagious** (m 1) | 180 (17.1%) | 169 (17.6%) | 11 (12.4%) | 1.566 | 0.133 |
| • No | 870 (82.9%) | 792 (82.4%) | 78 (87.6%) | | |
| • Yes | | | | | |
| **Cannabis Worsening COVID-19** (m 1) | 297 (28.3%) | 275 (28.6%) | 22 (24.7%) | 11.323 | **0.003** |
| • No | 752 (71.6%) | 686 (71.4%) | 66 (75.3%) | | |
| • Yes | | | | | |

m = missing values, N = sample size, T-Test = Student#39;s t-test, χ² = Chi-squared test.

The COVID-19 testing was negative in 91.4% (N = 961) of the participants and positive in 8.5% (N = 89). The average number of isolations was 0.74 (SD = 0.968) and 67 participants reported sharing cannabis during the pandemic (7.0%). Table 1 also shows the factors associated with the COVID-19 testing positive. COVID-19 testing positive was associated with higher academic performance and a higher number of isolations due to close contact with positive individuals.

Regarding the perception data associating the COVID-19 with cannabis use, 870 participants believed that cannabis smoke spreads the disease (82.9%), and 752 believed that cannabis use worsens COVID-19 (71.6%). COVID-19 testing positive was associated with a positive perception that cannabis use would worsen COVID-19.

## Cannabis use and estimated factors

From the total sample (Table 2), 669 participants (63.6%) were identified as non-cannabis users and 382 participants (36.4%) as cannabis users at some point in their lives. Over the last year, 887 participants (84.4%) were identified as

**Table 2. Sociodemographic, Psychological, and COVID-19 Variables Related to Cannabis Use in Adolescents.**

| Variables | LifeTime | | | Last Year | | | Last Month | | |
|---|---|---|---|---|---|---|---|---|---|
| | Non-Users N = 669 | Cannabis Users N = 382 | T-test o χ2. P valor | Non-Users N = 887 | Cannabis Users N = 164 | T-test o χ2. P valor | Non-Users N = 895 | Cannabis Users N = 156 | T-test o χ2. P valor |
| **COVID-19 testing** (m 1)<br>• No (negative)<br>• Yes (positive) | 619 (92.5%)<br>50 (7.5%) | 342 (89.8%)<br>39 (10.2%) | χ2 (2.388).<br>0.077 | 820 (92.4%)<br>67 (7.6%) | 141 (86.5%)<br>22 (13.5%) | χ2 (6.270).<br>**0.012** | 827 (92.5%)<br>67 (7.5%) | 133 (85.8%)<br>22 (14.2%) | χ2 (7.662).<br>**0.007** |
| *Sociodemographic and psychological* | | | | | | | | | |
| **Age** (14–18) (mean, SD)(m 9) | 15.21 (1.261) | 15.41 (1.330) | T (−2.436).<br>**0.015** | 15.30 (1.290) | 15.21 (1.288) | T (0.820).<br>0.410 | 15.30 (1.295) | 15.18 (1.256) | T (1.093).<br>0.275 |
| **Gender** (m 0) | | | | | | | | | |
| • Male | 309 (46.2%) | 175 (45.8%) | χ2 (0.014).<br>0.479 | 476 (53.7%) | 91 (55.5%) | χ2 (0.85).<br>0.366 | 413 (43.1%) | 71 (45.5%) | χ2 (0.021).<br>0.477 |
| • Female | 360 (53.8%) | 207 (54.2%) | | 411 (46.3%) | 73 (45.5%) | | 482 (53.9%) | 85 (54.5%) | |
| **Adolescents Educational Level** (m 0) | | | | | | | | | |
| • 10th Grade<br>• 11th Grade<br>• Vocational Training (VT) | 455 (68.0%)<br>184 (27.5%)<br>30 (4.5%) | 242 (63.4%)<br>118 (30.9%)<br>22 (5.8%) | χ2 (2.566).<br>0.277 | 579 (65.3%)<br>263 (29.7%)<br>45 (5.1%) | 118 (72.0%)<br>39 (23.8%)<br>7 (4.3%) | χ2 (2.771).<br>0.250 | 583 (65.1%)<br>266 (29.7%)<br>46 (5.1%) | 114 (73.1%)<br>36 (23.1%)<br>6 (3.8%) | χ2 (3.752).<br>0.153 |
| **Academic Performance** (mean, SD) (m 1) | 2.06 (0.934) | 2.18 (1.027) | T (−1.807).<br>0.071 | 2.08 (0.953) | 2.24 (1.050) | T (−1.915).<br>0.056 | 2.08 (0.953) | 2.26 (1.053) | T (−1.978).<br>**0.034** |
| **Self-esteem** (mean, SD)(m 1) | 28.73 (3.787) | 28.85 (4.187) | T (−0.485).<br>0.628 | 28.81 (3.901) | (4.123) | T (0.608).<br>0.543 | 28.80 (3.914) | 28.61 (4.068) | T (0.572).<br>0.567 |
| **Risk Profile of Cannabis Use** (mean, SD) (m 1) | | | | | | | | | |
| • Hopelessness | 21.04 (3.447) | 21.04 (3.444) | T (−0.013).<br>0.990 | 21.11 (3.468) | 20.67 (3.301) | T (1.485).<br>0.138 | 21.11 (3.465) | 20.63 (3.303) | T (1.609).<br>0.108 |
| • Impulsivity | 11.76 (3.142) | 12.02 (3.315) | T (−1.264).<br>0.206 | 11.72 (3.165) | 12.59 (3.342) | T (−3.192).<br>**0.001** | 11.73 (3.161) | 12.58 (3.378) | T (−3.063).<br>**0.002** |
| • Sensation Seeking | 16.20 (3.897) | 17.39 (3.981) | T (−4.746).<br>**<0.001** | 16.36 (3.969) | 18.09 (3.644) | T (−5.153).<br>**<0.001** | 16.38 (3.961) | 18.11 (3.685) | T (−5.083).<br>**<0.001** |
| • Anxiety | 12.055 (3.177) | 12.152 (3.312) | T (−0.468).<br>0.640 | 12.106 (3.260) | 12.006 (3.038) | T (0.363).<br>0.717 | 12.126 (3.267) | 11.884 (3.041) | T (0.864).<br>0.388 |
| *Behavioural* | | | | | | | | | |
| **Isolation's number** (mean, SD) (m 1) | 0.69 (0.947) | 0.83 (0.998) | T (−2.221).<br>0.27 | 0.71 (0.955) | 0.88 (1.027) | T (−2.076).<br>**0.038** | 0.71 (0.953) | 0.92 (1.035) | T (−1.564).<br>**0.0010** |
| **Sharing Cannabis in Pandemic** (m 158) | | | | | | | | | |
| • No | 740 (82.86%) | 153 (17.13%) | χ2 (102.87) | 846 (94.74%) | 47 (52.6%) | χ2 (309.91) | 797 (89.24%) | 96 (10.76%) | χ2 (275.128) |
| • Yes | 5 (7.46%) | 62 (92.53%) | **<0.001** | 25 (37.31%) | 42 (62.68%) | **<0.001** | 8 (11.94%) | 59 (88.06%) | **<0.001** |

*(Continued)*

**Table 2.** (Continued)

| Variables | LifeTime | | | Last Year | | | Last Month | | |
|---|---|---|---|---|---|---|---|---|---|
| | Non-Users N=669 | Cannabis Users N=382 | T-test o χ2. P valor | Non-Users N=887 | Cannabis Users N=164 | T-test o χ2. P valor | Non-Users N=895 | Cannabis Users N=156 | T-test o χ2. P valor |
| *Perceptions* | | | | | | | | | |
| **Cannabis' Smoke Contagious** (m 1) | | | | | | | | | |
| • No | 101 (15.1%) | 79 (20.7%) | χ2 (5.432). | 136 (15.3%) | 44 (27%) | χ2 (13.183). | 137 (15.3%) | 43 (27.7%) | χ2 (14.382). |
| • Yes | 568 (84.9%) | 302 (79.3%) | **0.013** | 751 (84.7%) | 119 (73%) | **<0.001** | 758 (84.7%) | 112 (72.3%) | **<0.001** |
| **Cannabis Worsening COVID-19** (m 1) | | | | | | | | | |
| • No | 150 (22.4%) | 147 (38.6%) | χ2 (31.674). | 211 (23.8%) | 86 (52.8%) | χ2 (57.060). | 214 (23.9%) | 83 (53.5%) | χ2 (57.284). |
| • Yes | 518 (77.4%) | 234 (61.4%) | **<0.001** | 675 (76.1%) | 77 (47.2%) | **<0.001** | 680 (76.0%) | 72 (46.5%) | **<0.001** |

m=missing values, N=sample size, T-Test=Student#39;s t-test, χ²=Chi-squared test.

non-cannabis users and 164 participants (15.6%) as cannabis users. In the last month, 895 participants (85.2%) were identified as non-cannabis users and 156 participants (14.8%) as cannabis users.

Among the sociodemographic and psychological variables, the only variable that was significantly associated with cannabis use across all three periods was the personality trait of sensation seeking, which had a higher mean in users compared to non-users. For cannabis use in the last month and the last year, a positive association was also found with the trait of impulsivity. Academic performance was associated with cannabis use in the last month, interestingly showing higher performance among users. Age was significantly positively associated with lifetime cannabis use.

COVID-19 testing positive was significantly associated with cannabis use in the last year, last month, and marginally with lifetime use. In all instances, the risk of COVID-19 testing positive was higher in users compared to non-users. Additionally, perceptions such as "cannabis smoke transmits COVID-19" and "cannabis use worsens COVID-19" were correlated with the three frequencies of cannabis use: lifetime, last year and last month. For instance, cannabis users showed a slightly lower perception that cannabis' smoke could spread COVID-19 compared to non-users. However, non-cannabis users reported having favorable perceptions about cannabis use could worsen COVID-19 compared to users, who reported to have the perception that cannabis could not worsen COVID-19. Finally, the mean number of isolations was higher among users in the last year and the last month.

### Relationship of COVID-19 testing positive and Cannabis Use

The Odds Ratio (OR) was calculated for all three period of time, being respectively: 1.9096, 95% CI=1.1425 to 3.1918 for lifetime cannabis use, 2.0417, 95% CI=1.2196 to 3.4180 for last year cannabis use, and 1.4118, 95% CI=0.9101 to 2.1900 for last month cannabis use.

In the regression analysis, sharing cannabis during the pandemic and not having the perception about cannabis worsens COVID-19, were related to cannabis use in all the explored frequencies of cannabis use (lifetime, last year and last month). For last month cannabis use, a COVID-19 testing positive was also associated with cannabis use (OR=2.898). Nagelkerke#39;s indicator of explained variance was 40% for last month cannabis use (Table 3).

### Discussion

This study is the first conducted in Andalusia to explore the estimated association between reporting COVID-19 testing positive and using cannabis among Andalusian adolescents aged 14–18, across three different cannabis use' periods:

**Table 3. Resultant Variables in Logistic Regression Analysis for Cannabis Consumption on Adolescents.**

| Variables | Lifetime | | | | Last Year | | | | Last Month | | | |
|---|---|---|---|---|---|---|---|---|---|---|---|---|
| | p | OR | IC 95% | | p | OR | IC 95% | | p | OR | IC 95% | |
| | | | Inf | Sup | | | Inf | Sup | | | Inf | Sup |
| Testing COVID-19 (yes VS no) | .206 | 1.390 | .834 | 2.317 | .325 | 1.581 | .636 | 3.932 | **.001** | 2.898 | 1.525 | 5.507 |
| *Behavioural* | | | | | | | | | | | | |
| Number Isolations | .778 | 1.014 | .918 | 1.121 | .961 | .996 | .835 | 1.187 | .962 | 1.003 | .870 | 1.158 |
| Share Cannabis During Pandemic (no VS yes) | **<.001** | 51.688 | 12.420 | 215.098 | **<.001** | 21.663 | 11.388 | 41.208 | **<.001** | 64.302 | 27.033 | 152.949 |
| *Perceptions* | | | | | | | | | | | | |
| Cannabis' Smoke contagious COVID-19 (no Vs yes) | .709 | 1.082 | .717 | 1.632 | .311 | .720 | .381 | 1.360 | .425 | .800 | .462 | 1.385 |
| Cannabis Worsens COVID-19 (no VS yes) | **<.001** | .538 | .382 | .757 | **.001** | .390 | .220 | .691 | **<.001** | .386 | .241 | .616 |
| **Chi-square omnibus** | 164.121 (<.001) | | | | 173.603 (<.001) | | | | 256.995 (<.001) | | | |
| **Nagelkerke R²** | .216 | | | | .363 | | | | .404 | | | |

95% IC = 95% confidence interval; OR= odds ratio; p = p value. The analyses were controlled for sociodemographic and psychological variables.

lifetime use, use in the last year, and using in the last month. Additionally, this article addresses one of the knowledge gaps identified by Torrejón-Guirado in previous studies: the pandemic situation and its relationship with cannabis use [26].

Regarding the prevalence of cannabis use, our results showed that lifetime cannabis use was higher among females than males, while in ESTUDES 2023 showed that lifetime cannabis use was higher among males [10]. Furthermore, looking at the last month cannabis use, while our results showed similar outcomes to those observed over a lifetime cannabis use (i.e., higher consumption in females), ESTUDES survey indicated that the prevalence of cannabis use was nearly twice as high among males (1.7%) compared to females (1%) [10].

Regarding the estimated association between reporting COVID-19 testing positive and using cannabis, the OR for cannabis use in the last month and COVID-19 testing positive was 2.898, suggesting that individuals diagnosed with COVID-19 are almost three times more likely to have used cannabis in the last month. Yet, the Nagelkerke R² values suggest that 60% of the variability is attributed to factors not accounted for in the model, especially in the case of lifetime and last-year cannabis use. Thus, caution is requested in the interpretation of the data, as well as when interpreting the confidence coefficients (some intervals were wide, reducing the statistical significance). Nevertheless, our finding is in line with a study which reflected that 52% of those students (aged 18–26) who smoked electronic cigarettes and cannabis were 1.85 times more likely to be diagnosed with COVID-19 [16]. Additionally, the fear of COVID-19 testing positive has been linked to increased cannabis use [27].

Regarding the perception of "cannabis worsens COVID-19", our finding suggests on one hand, that this perception is slightly higher in those who have had COVID-19 (75.3%) compared to those who have not had it (71.4%), and on the other hand, individuals who do not endorse this belief are more likely to use cannabis. However, no research has previously studied the associations between beliefs about the impact of cannabis use on this infectious disease. Although our study cannot conclude that cannabis use worsens COVID symptoms, a study (n = 13,099) of adults showed that the frequency of cannabis use could be considered a possible predictor of COVID-19 mortality risk [28]. So, giving the relevance of this finding, it would be beneficial to conduct studies in adolescent population to implement health recovery and prevention measures. These measures could include informational sessions for adolescents about the consequences of cannabis use on COVID-19 testing positive, highlighting the increased probability of developing respiratory issues such as severe pneumonia, and basing also on relevant guidelines such as Parrado-Gónzalez et al., mainly, as far as Andalusia

and Spain are concerned [29]. Moreover, a retrospective study found that adult users of substances such as cannabis and tobacco are significantly more vulnerable to COVID-19 and its complications, making them more likely to require hospitalization and face and increased risk of death compared to non-users [30]. We recommend investigating a possible causal association between a low perception that cannabis worsens COVID-19 and the presence of the disease. As low-risk perception has been widely studied in the literature as a relevant in the adoption of risky behaviours, it may also impact the acquisition/progression of the disease. This highlights the importance of addressing such beliefs through health education.

Educational interventions could play a relevant role in preparing the population for future pandemics by anticipating the consequences of COVID-19, particularly for individuals with pre-existing conditions affecting the respiratory system. Some of these measures could include sharing information through adolescent and young adult consumers who have experienced hospitalization and suffered significant health effects due to the progression of the disease after being diagnosed. Moreover, misconceptions about cannabis and COVID-19 may arise from incomplete information, such as media reports suggesting a potential protective effect or the known anti-inflammatory properties of cannabidiol (CBD) [31]. Some adolescents might wrongly assume that cannabis consumption enhances immunity or mitigates severe inflammatory responses associated with COVID-19. These misinterpretations highlight the need for clear, evidence-based education on cannabis and its actual health effects.

In relation to the higher frequency of isolations among cannabis users, no studies have been identified that directly analyse the relationship between the number of isolations and the likelihood of testing a positive COVID-19, while several studies have suggested that individuals who self-isolated during the COVID-19 pandemic might have increased their cannabis consumption due to the emotional distress caused by isolation [32–34]. Cannabis use may have risen during isolations due to factors such as stress, anxiety, depression, and boredom. However, a more thorough understanding of this reality should be pursued, with questions that allow for a broader exploration of these factors, since it is also possible that individuals who use cannabis may have more positive test results and, consequently, more isolations. Although the interaction has not been studied, this interpretation is plausible and could occur, while always considering the limitations of our exploratory analysis.

Regarding the correlation of COVID-19 testing positive and having a good academic performance, similar to our results, a study conducted in China (n = 1316) with high school students found that, despite isolation due to COVID-19, the students maintained high academic performance [35]. This could be because these studies were conducted approximately one year after the state of emergency. Therefore, educational institutions prepared themselves and provided their students with tools to ensure their academic performance. Despite that isolation encompasses numerous barriers to potential academic development, such as difficulties in accessing adequate computing and technology, distracting home environments, and the socio-emotional cost of isolation [32,36,37]. This estimated association would therefore need to be analysed in depth, for instance, through longitudinal studies.

Finally, interesting correlations were found between cannabis use and certain behavioural (sharing cannabis during the pandemic and isolations) and perceptions variables related to COVID-19 (cannabis smoke contagion COVID-19 and cannabis worsening COVID-19) in bivariate and multivariate analysis. For instance, concerning the perceptions of "cannabis smoke contagion COVID-19", no published studies were found that examined the possible association between the perception that "cannabis smoke could transmit COVID-19" and using cannabis, either than COVID-19 testing. However, in the context of tobacco, tobacco smoke has been shown to facilitate COVID-19 transmission [38,39]. To prevent the spread of COVID-19, it may be relevant for adolescents to understand the harmful consequences of group cannabis use during COVID-19 outbreaks in their environments.

At last, it should be considered that as the adolescents' responses are based on perceptions rather than experiences, it should also be considered that cannabis users may respond in a socially desirable manner, aligning with perceived health norms to minimize potential health harm. Nevertheless, and given the estimated association found, it is important that educators and nurses explain to them that the consequences of having COVID-19 while using cannabis

could be detrimental for health. It would also be relevant to study the differences between the effects of CBD and THC separately to obtain clearer conclusions, as some existing literature suggests that CBD could be a protective factor against COVID-19 [40].

### Strengths and limitations

One limitation of this study is its cross-sectional design. We recommend conducting longitudinal studies to establish causal relationships regarding cannabis use through these factors. Furthermore, the study#39;s reliance on a limited sample of participants who tested positive for COVID-19 (N = 89) restricts the statistical power and generalizability of the findings. A second limitation is that, due to the contemporary nature of the study topic, the questions related to COVID-19 have not been previously validated (potential recall bias or tic errors). It would thus be advisable to design and validate a tool to measure COVID-19-related variables in relation to cannabis consumption. Given that this study occurred during the pandemic with unusual measures in place, most secondary schools did not allow in-person studies, preventing close interaction with students. The third limitation is related to the Rosenberg scale. Although it is a previously validated instrument, our study found a Cronbach#39;s alpha of 0.459. Yet, no studies have been found reporting Cronbach's alpha as low as 0.5 when applied to adolescents. Such a low reliability coefficient could indicate issues related to scale administration, item comprehension, or cultural adequacy of the instrument. It is possible that the low Cronbach's alpha is influenced by adolescents' self-perception of their own self-esteem. Given the sensitive and personal nature of self-esteem, adolescents may struggle to respond honestly, particularly in contexts where expressing vulnerability is socially discouraged. Furthermore, difficulties in instrument administration, such as lack of privacy or social pressure during responses, may also contribute to lower reliability estimates.

Additionally, although the regression analysis controls for sociodemographic and psychological variables, there may be other unmeasured confounding factors influencing the results (e.g., the social environment in which cannabis use occurs, which could significantly impact both COVID-19 risk, or other factors such as cannabis use patterns, cannabis' products, family dynamics or mental health). In fact, the multivariate analysis suggests that 60% of the variability is attributed to factors not accounted for in the model, especially in the case of lifetime and last-year cannabis use. Thus, caution is requested in the interpretation of the data, as well as when interpreting the confidence coefficients (some intervals were wide, reducing the statistical significance).

It should also be considered that the results of this study should be interpreted with caution, as it is based on a self-reported survey (i.e., reporting bias), where data on the specific types of cannabis products (such as herb, resin, CBD) and methods of consumption (such as vaping or oral intake) were not specified. Moreover, in Spain, as in other countries, there is a partial regulation of cannabis supplies. For instance, low-dronabinol (delta-9-THC) cannabis products are available, which may contribute to a reduced risk perception of cannabis use among youth [35]. Furthermore, considering these trends, new therapeutic approaches focusing on harm reduction are increasingly being adopted by health practitioners and should be considered. Future research could explore these variables to provide a more comprehensive understanding of the relationship between cannabis use and COVID-19 outcomes, as well as potential strategies to mitigate the impact of substance use on infection rates. They also could consider reclassifying participants into mutually exclusive groups (no use = 669; last-month use = 156; use more than one month ago = 226) and conducting supplementary multinomial logistic regression analyses.

Finally, a selection bias is that the recruited participants were school adolescents. It would be recommended to include adolescents who not attend school since they may have different characteristics. A last limitation is the variable "sharing cannabis during the pandemic" should be also interpreted with caution, as the self-administered survey did not specify whether it referred to just a single puff, an entire joint, or other details such as a cannabis-containing cigarette. Future studies aiming to investigate the relationship between COVID-19 and shared cannabis use, in order to determine whether it increases transmission, should take these considerations into account.

The primary strength of this study is that no other studies in Spain have been found to relate cannabis consumption with COVID-19 variables, such as COVID-19 testing positive. Diagnosing COVID-19 in people who use cannabis helps protect respiratory and immune health, allows for better symptom and treatment management, and reduces the risk of complications. Another strength is the heterogeneity of the sample and its sufficient size. Different frequencies for measuring cannabis consumption were also included, allowing for a closer approximation to the reality of this context in Andalusia. A final strength of this study is that, despite the small sample size of participants who reported testing positive for COVID-19, significant cross-sectional estimated associations were found with two variables.

## Conclusion

The findings of this exploratory study provide additional context-specific data regarding cannabis use among adolescents in Western Andalusia during the COVID-19 pandemic. An estimated association was observed between COVID-19 positivity and cannabis use, particularly last-month consumption where the odds was almost three points higher. Moreover, cannabis users showed a slightly lower perception that cannabis' smoke could spread COVID-19 while non-cannabis users reported having favorable perceptions about cannabis use could worsen COVID-19. This perception is also higher in those that were positive testing COVID-19. Finally, the mean number of isolations was higher among cannabis users and positive testing. As consuming cannabis has been identified as a predictor of COVID-19 hospitalization and mortality risk we think that our study gives some context-insights that could be relevant for preparing for future public health emergencies and for informed future large scale research. It is also important of leverage the conscientious about the effect of cannabis use among adolescents and their context, which can be exacerbated in a pandemic situation.

## Author contributions

**Conceptualization:** María-Carmen Torrejón-Guirado, Miguel Ángel Baena-Jiménez, Marta Lima-Serrano.

**Data curation:** María-Carmen Torrejón-Guirado, Miguel Ángel Baena-Jiménez.

**Formal analysis:** María-Carmen Torrejón-Guirado.

**Funding acquisition:** Marta Lima-Serrano.

**Investigation:** Miguel Ángel Baena-Jiménez, Marta Lima-Serrano.

**Methodology:** María-Carmen Torrejón-Guirado, Miguel Ángel Baena-Jiménez, Marta Lima-Serrano.

**Project administration:** María-Carmen Torrejón-Guirado.

**Software:** María-Carmen Torrejón-Guirado.

**Supervision:** Marta Lima-Serrano.

**Writing – original draft:** María-Carmen Torrejón-Guirado.

**Writing – review & editing:** Miguel Ángel Baena-Jiménez, Marta Lima-Serrano.

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
