## [Decision Letter · Decision Letter 0]

7 Jan 2025

PONE-D-24-49229COVID-19 and cannabis use in students from 14 to 18 years old: addressing its related factors.PLOS ONE

Dear Dr. Miguel Ángel Baena-Jiménez

Thank you for submitting your manuscript to PLOS ONE. After careful consideration, we feel that it has merit but does not fully meet PLOS ONE’s publication criteria as it currently stands. Therefore, we invite you to submit a revised version of the manuscript that addresses the points raised during the review process.

Please address every point raised by both the reviewers, with detailed explanations for each point, especially by Reviewer 2, as I agree with almost all of their suggestions.

We look forward to receiving your revised manuscript.

Kind regards,

Aman Goyal

Academic Editor

PLOS ONE

Journal Requirements:

“This work was supported by the University of Seville research program (VPPI-US) in terms of a pre-doctoral contract of Torrejón-Guirado.”

4. In this instance it seems there may be acceptable restrictions in place that prevent the public sharing of your minimal data. However, in line with our goal of ensuring long-term data availability to all interested researchers, PLOS’ Data Policy states that authors cannot be the sole named individuals responsible for ensuring data access (http://journals.plos.org/plosone/s/data-availability#loc-acceptable-data-sharing-methods).

Additional Editor Comments:

Please address every point raised by both the reviewers, with detailed explanations for each point, especially by Reviewer 2, as I agree with almost all of their suggestions.

Reviewers' comments:

Reviewer's Responses to Questions

**Comments to the Author**

1. Is the manuscript technically sound, and do the data support the conclusions?

Reviewer #1: Yes

Reviewer #2: Partly

2. Has the statistical analysis been performed appropriately and rigorously? 

Reviewer #1: Yes

Reviewer #2: No

3. Have the authors made all data underlying the findings in their manuscript fully available?

Reviewer #1: Yes

Reviewer #2: No

4. Is the manuscript presented in an intelligible fashion and written in standard English?

Reviewer #1: Yes

Reviewer #2: No

5. Review Comments to the Author

Reviewer #1: -Research Summary and General Impression

The study examines sociodemographic, psychological, and behavioral variables related to cannabis use and COVID-19 diagnosis in adolescents. While the results are interesting, the cross-sectional design limits causal conclusions.

-Strengths:

Real-world observational study providing insights into behaviors during the pandemic.

Considerable and diverse sample size.

-Major Issues:

COVID-19 Diagnosis Validation: Lack of diagnostic testing may affect result validity.

Causality: Avoid claims of “strong association” and emphasize that the study generates hypotheses rather than establishing causation.

-Minor Issues:

Descriptive Statistics: Include detailed age distribution and other sociodemographic variables.

STROBE Checklist: The article mentions a flowchart but omits the full checklist.

-The study presents results from primary scientific research:

The study is a cross-sectional observational study (survey), suitable for generating hypotheses. Clarify the study type in the methods section to prevent misunderstandings regarding causality.

-The reported results have not been published elsewhere:

It is unclear if the results are novel. The authors should explicitly if the findings have not been submitted elsewhere to avoid duplication.

-Experiments, statistics, and other analyses are conducted to a high standard and described in sufficient detail:

Number of Participants: Adolescents aged 14-18 were included. Specify how many participants were under 18 and clarify how parental informed consent (IC) was obtained. Provide details on "passive consent" to ensure ethical compliance.

Descriptive Analysis: Only the median age is given. Include a percentage distribution of ages for a more comprehensive sample understanding.

-Conclusions are appropriately presented and supported by data:

COVID-19 Diagnosis: Diagnosis was based on a binary question ("Yes/No"), with no confirmatory testing. Discuss the limitations of this method and potential recall bias or diagnostic errors.

Association vs. Causation: The study suggests an association between COVID-19 diagnosis and cannabis use, but cross-sectional design limits causal inference. Replace terms like "association" with "estimate" or "hypothesis generation" to reflect the study's nature.

Severity of COVID-19: Consider including information on the severity of COVID-19 cases among participants.

-The article is intelligible and written in standard English:

The article is well-written. Consistently replace “association” with “estimate” or “hypothesis generation” where appropriate.

-The research meets ethical standards and research integrity:

Minor Consent: For ages 14-18, clarify how parental informed consent was obtained. Confirm if all minors had signed consent from guardians.

-The article adheres to appropriate reporting guidelines and community standards for data availability.

STROBE Checklist: The article mentions a flowchart but omits the full checklist.

-Requests:

1.STROBE checklist to ensure all relevant observational study items are addressed.

2.Confirm if 100% of parental or legal guardian consents were obtained for minors.

3.Suggest more detailed statistical descriptions of the sample, particularly age distribution.

4.Replace terms like "association" with "estimate" or "hypothesis generation" to reflect the study's nature.

5.COVID-19 Diagnosis Validation: Lack of diagnostic testing may affect result validity. Discuss the limitations of this method and potential recall bias or diagnostic errors.

Reviewer #2: The full review exceeds 20,000 characters, and is placed in attachement.

Conclusion:

"[…] In conclusion, while the study explores a relevant and timely topic, it fails to meet the necessary standards of scientific rigour for publication in PLOS ONE as such. The manuscript's primary limitations include:

- The use of outdated epidemiological data, specifically from the 2018 Spanish National Plan on Drugs survey (ESTUDES), neglecting more recent and relevant studies such as the 2023 ESTUDES report and the 2024 EDADES report. This failure to incorporate the most current data undermines the study's contextualisation and relevance.

- A lack of precision and clarity in the formulation of the research questions and the interpretation of results, for example, using the term "students" when "adolescents" may be more appropriate, the poorly defined hypothesis, and ambiguous wording concerning cannabis sharing.

- An over-reliance on a small sample of participants who tested positive for COVID-19 (N=89), which limits the statistical significance and generalizability of the findings.

- A problematic mixing of objective and subjective data, most evident in Table 1, which makes interpretation difficult.

- The lack of a thorough discussion of harm reduction strategies, which have been proven to be effective and should be considered when addressing youth cannabis use

- A tendency to draw strong conclusions from weak associations, such as the suggestion of links between school absenteeism and COVID-19.

Overall, the manuscript needs substantial revision to address these critical issues. A more robust methodology, clearer research questions, up-to-date data, and a more nuanced discussion of the findings are required before this study can be considered a valid contribution to the scientific record. Without these fundamental improvements, the conclusions drawn from the data remain questionable."

6. PLOS authors have the option to publish the peer review history of their article (what does this mean?). If published, this will include your full peer review and any attached files.

Reviewer #1: **Yes:**Cláudio Calixto Carlos da Silva

Reviewer #2: No

---

## [Author Response · Author response to Decision Letter 1]

1 Apr 2025

Dear reviewers,

We have uploaded 4 new documents: the revised manuscript with tracks and without tracks changes, and other two documents answering the reviewers comments (they are named as: "answer to reviewer 1" and "answer to reviewer 2").

We appreciate your valuable comments to achieve and to improve the quality of the manuscript.

Best regards

Miguel Ángel Baena Jiménez

---

## [Decision Letter · Decision Letter 1]

17 Sep 2025

PONE-D-24-49229R1Exploring the Relationships Between Reporting Testing Positive for COVID-19 and Cannabis Use: an Exploratory Study.PLOS ONE

Dear Dr. Baena-Jiménez,

Thank you for submitting your manuscript to PLOS ONE. After careful consideration, we feel that it has merit but does not fully meet PLOS ONE’s publication criteria as it currently stands. Therefore, we invite you to submit a revised version of the manuscript that addresses the points raised during the review process.

We look forward to receiving your revised manuscript.

Kind regards,

David Adedia, Ph.D

Academic Editor

PLOS ONE

Journal Requirements:

Additional Editor Comments :

The authors should consider the following:

The Cronbach's alpha of 0.459, is too low. The recoding process should be checked to ensure the low value is not due to that. In addition, the authors should consider assessing the validity of the scale as well as improve its reliability. A low reliability defeats the findings.

When the prevalence is either far above or below 0.5, the logistic regression does not provide efficient results. The prevalences per the different times is low and complementary log-log model would have been better.

In Table 3, IC should be changed to CI

P valor should be written as p-value.

T-test o χ2 should be written properly.

The document should be proofread to correct grammatical and punctuation errors, especially where spaces are required.

Reviewers' comments:

Reviewer's Responses to Questions

**Comments to the Author**

1. If the authors have adequately addressed your comments raised in a previous round of review and you feel that this manuscript is now acceptable for publication, you may indicate that here to bypass the “Comments to the Author” section, enter your conflict of interest statement in the “Confidential to Editor” section, and submit your "Accept" recommendation.

Reviewer #1: All comments have been addressed

Reviewer #3: (No Response)

Reviewer #4: (No Response)

2. Is the manuscript technically sound, and do the data support the conclusions?

Reviewer #1: Yes

Reviewer #3: Yes

Reviewer #4: Yes

3. Has the statistical analysis been performed appropriately and rigorously? 

Reviewer #1: Yes

Reviewer #3: Yes

Reviewer #4: No

4. Have the authors made all data underlying the findings in their manuscript fully available?

Reviewer #1: Yes

Reviewer #3: Yes

Reviewer #4: Yes

5. Is the manuscript presented in an intelligible fashion and written in standard English?

Reviewer #1: Yes

Reviewer #3: Yes

Reviewer #4: Yes

6. Review Comments to the Author

Reviewer #1: Reviewer Comments (Round 2):

Thank you for the revised version of the manuscript. The authors have addressed most of the points raised in the initial review, and I appreciate the effort made to improve the clarity and transparency of the study. The inclusion of the STROBE checklist, detailed age distribution, and clarification regarding consent procedures are all welcome additions.

However, two important concerns remain and need to be addressed before the manuscript is suitable for publication:

Causal Language in the Conclusion and Interpretation of Results:

1.Despite the authors’ efforts to replace terms such as "association" with "estimate" or "correlation" in some parts of the manuscript, the conclusion still uses language that implies causality, which is not appropriate for a cross-sectional observational study. For instance, the statement:

“COVID-19 testing positive was associated with cannabis use, particularly last month consumption. It was also associated with increased academic performance, a higher number of isolations, and the perception that cannabis consumption worsens COVID-19.”

should be revised to avoid causal implications. This phrasing suggests a level of certainty and directionality that the study design does not support. I recommend using more cautious language such as: “An estimated association was observed between…” or “Data showed a correlation between…”

2.Introduction – Misleading Interpretation of the Literature:

The manuscript references the following article to support the claim that cannabis use directly increases the risk of acquiring COVID-19 (pag 3)

Griffith, N. B., Baker, T. B., Heiden, B. T., Smock, N., Pham, G., Chen, J., ... & Chen, L. S. Cannabis, Tobacco Use, and COVID-19 Outcomes. JAMA Network Open 7.6 (2024): e2417977-e2417977.

However, this cited study supports the broad assertion that “cannabis use was associated with an increased risk of hospitalization and ICU admission.” The authors of the cited study provide a more nuanced interpretation of the data, and this should be accurately reflected in the manuscript. Please revise this sentence to ensure it is a faithful representation of the original source, and avoid overstating the evidence. A more precise and referenced summary of the findings from Griffith et al. should be included.

Overall, the manuscript has improved significantly, and I thank the authors for their thoughtful revisions. I encourage them to address the remaining issues to ensure the manuscript reflects the study’s methodological limitations and the referenced literature with appropriate caution.

Reviewer #3: I have carefully reviewed this manuscript and below is my decision.

The paper is well-suited for publication in the Plos One. All comments are well-improved the quality of paper. Good luck!

Reviewer #4: The authors sought to explore the relationship between cannabis use and COVID-19 test results, adjusting for behavioural, psychological and sociodemographic variables. They basically utilized a cross-sectional sampling methodology with validated instruments to assess the key variables.

Cannabis use was defined based on recency into lifetime use, last year or last month and the authors used chi square and logistic regressions to explore the relationship.

In short, to address the hypothesis tables 2 and 3 are necessary but table 1 is unnecessary. For table 3 more details on the model-building strategies are needed and perhaps for each classification of cannabis use the authors may wish to present a parsimonious model. Another consideration that the authors could explore is dividing cannabis use into mutually exclusive groups of no use =669, last month 156 and some use more than a month ago (226) and use multinominal logistic regression. Additionally, the 95% CI for the Odds ratio would be a more useful statistic to include in Table 3.

The introduction and discussion covers the salient points.

7. PLOS authors have the option to publish the peer review history of their article (what does this mean?). If published, this will include your full peer review and any attached files.

Reviewer #1: **Yes:**Cláudio Calixto Carlos da Silva

Reviewer #3: No

Reviewer #4: **Yes:**Marvin Reid

---

## [Author Response · Author response to Decision Letter 2]

21 Oct 2025

REVIEWER 1

We sincerely thank Reviewer #1 for their thoughtful and constructive feedback on the revised version of our manuscript. We appreciate the positive remarks regarding the improvements made, including the inclusion of the STROBE checklist, detailed age distribution, and clarification of the consent procedures. Below, we address each of the remaining concerns point by point.

1. Causal Language in the Conclusion and Interpretation of Results

Reviewer’s comment:

The reviewer correctly notes that some statements in the conclusion still imply causality, which is not appropriate for a cross-sectional design. The reviewer suggests using more cautious phrasing such as “An estimated association was observed between…” or “Data showed a correlation between…”.

Response:

We fully agree with the reviewer’s observation and have revised the relevant sections of the manuscript to ensure that all interpretations are expressed in non-causal terms. Specifically, the statement in the conclusion now reads:

“An estimated association was observed between COVID-19 positivity and cannabis use, particularly last-month consumption. Data also showed correlations with higher academic performance, a greater number of isolations, and the perception that cannabis consumption worsens COVID-19 outcomes.”

We have carefully reviewed the entire manuscript to ensure consistent use of language that accurately reflects the cross-sectional nature of the study and avoids any implication of causality.

2. Introduction – Misleading Interpretation of the Literature

Reviewer’s comment:

The reviewer points out that our citation of Griffith et al. (2024) could be interpreted as overstating the evidence, as the cited study reports associations with hospitalization and ICU admission rather than directly with infection risk.

Response:

We thank the reviewer for this valuable clarification. We have revised the text in the introduction to more accurately reflect the findings of Griffith et al. (2024). The revised sentence now reads:

“Previous studies, such as that by Griffith et al. (2024), have reported that cannabis use was associated with an increased risk of hospitalization and ICU admission among individuals with COVID-19.”

We believe this modification provides a more faithful representation of the original study and avoids overstating its conclusions. The corresponding reference has been retained and properly contextualized within our discussion.

REVIEWER 3

We really appreciate the comments of the reviewer.

REVIEWER 4

We sincerely thank Reviewer #4 for their helpful and constructive feedback, which has guided further refinement of our statistical analysis and presentation.

Comment:

The reviewer suggests providing more details on the model-building strategies used for Table 3, considering parsimonious models for each cannabis use category, exploring mutually exclusive cannabis-use groups with multinomial logistic regression, and including 95% confidence intervals for the odds ratios.

Response:

We appreciate these valuable suggestions:

1. Model-Building Strategy and Parsimonious Models and Cannabis Use Categories:

We appreciate your suggestion; however, this was not the objective of our study. Our aim was not to identify the most parsimonious model, but rather to conduct an exploration analysis by retaining all variables in order to examine whether any associations existed. Nevertheless, we have included in the limitation section that in future studies that consideration should be taking into account. It is reflected as follows:

“They also could consider reclassifying participants into mutually exclusive groups (no use = 669; last-month use = 156; use more than one month ago = 226) and conducting supplementary multinomial logistic regression analyses.”

2. 95% Confidence Intervals:

We have revised Table 3 and the 95% confidence intervals has been already included for all reported odds ratios, as suggested. We agree that this addition substantially enhances the interpretability and statistical transparency of the results.

We thank the reviewer for these insightful recommendations, which have strengthened both the methodological rigor and clarity of our results.

---

## [Decision Letter · Decision Letter 2]

29 Jan 2026

PONE-D-24-49229R2Exploring the Relationships Between Reporting Testing Positive for COVID-19 and Cannabis Use: an Exploratory Study.PLOS One

Dear Dr. Baena-Jiménez,

Thank you for submitting your manuscript to PLOS ONE. After careful consideration, we feel that it has merit but does not fully meet PLOS ONE’s publication criteria as it currently stands. Therefore, we invite you to submit a revised version of the manuscript that addresses the points raised during the review process.

The authors did an admirable job addressing a slate of critiques from multiple reviewers over two rounds of reviews. However, there are a few gaps that remain, as pointed out by Reviewer 2. It is unfortunate that this reviewer did not get a chance to comment on the first revision, but after their careful assessment of Revision 2, the critiques listed in their current review are valid and need addressing. I understand, as Reviewer 2 does, potential frustration that this additional round of reviews may bring. However, I also agree with Reviewer 2 that with these few remaining adjustments, the paper will be publishable and high-quality. Along with Reviewer 2, I apologize for the extended review process. I look forward to receiving your final round of revisions.

We look forward to receiving your revised manuscript.

Kind regards,

Magdalena Szaflarski, PhD

Academic Editor

PLOS One

Journal Requirements:

Reviewers' comments:

Reviewer's Responses to Questions

**Comments to the Author**

1. If the authors have adequately addressed your comments raised in a previous round of review and you feel that this manuscript is now acceptable for publication, you may indicate that here to bypass the “Comments to the Author” section, enter your conflict of interest statement in the “Confidential to Editor” section, and submit your "Accept" recommendation.

Reviewer #1: All comments have been addressed

Reviewer #2: (No Response)

Reviewer #3: All comments have been addressed

2. Is the manuscript technically sound, and do the data support the conclusions?

Reviewer #1: Yes

Reviewer #2: Partly

Reviewer #3: Yes

3. Has the statistical analysis been performed appropriately and rigorously? 

Reviewer #1: Yes

Reviewer #2: N/A

Reviewer #3: Yes

4. Have the authors made all data underlying the findings in their manuscript fully available?

Reviewer #1: Yes

Reviewer #2: Yes

Reviewer #3: Yes

5. Is the manuscript presented in an intelligible fashion and written in standard English?

Reviewer #1: Yes

Reviewer #2: No

Reviewer #3: Yes

6. Review Comments to the Author

Reviewer #1: I would like to thank the authors for the substantial improvements made throughout the revision process. The manuscript is now clearer, more consistent, and the main concerns raised in previous rounds have been adequately addressed. The adjustments to causal language, clarification of the introduction, and revisions throughout the text have strengthened the overall quality of the work.

At this point, I believe the manuscript is suitable for publication pending only minor formatting adjustments. Several tables still present formatting inconsistencies, including:

Table 1 appearing split across two pages;

Placeholder text such as “insertheretable1”, “insertheretable2”, and “ insertheretable3” remaining in the manuscript.

I kindly recommend a final formatting check to ensure that all tables appear as single, complete blocks and that placeholders are removed. These adjustments are minor and do not affect the scientific content of the manuscript.

Reviewer #2: I thank the authors for their work and for their involvement in the review process. I also take the opportunity to wish them all an happy new year.

I note the authors have "carefully reviewed the entire manuscript to ensure consistent use of language that accurately reflects the cross-sectional nature of the study and avoids any implication of causality." While this should not have required two rounds of peer-review, it is indeed finally appropriate in the body of the manuscript; it is, however not, in the Abstract. My review focuses therefore on the Abstract, as well as the introduction.

An outstanding element that deserve address is the apparent deterritorialisation of the results, which does not seem justified. This study was conducted in four specific urban contexts of Western Andalucía, and among adolescents. None of these elements are present in the title (geographical location [Andalucia], type of environment [urban], population studied [scholarised adolescents]). It is critical that these elements appear in the title, and clearly articulated in the abstract.

The importance of accurately representing the specificities of the localised regional context within which the study was conducted (urban context in Western Andalucía) is key.

I also stress that the territory in question (four specific large cities in Western Andalucía) is located at the najor entry point of cannabis resin for illicit markets, fromNorthern Africa to Europe. This absolutely unique situation is nowhere mentioned. Other geographical, socio-cultural, and political contextual elements may have been welcome.

Furthermore, I note that imported cannabis drugs from that Northern African region often constitute “hashish” (cannabis resin) and not “flowers” (dry herbal tops) which are the subject of most published research. Besides possible differences in effects on individual and public health, the consumption of hashish is, I believe, directly associated with that of tobacco. I may have missed it, but this critical element was not mentioned in the manuscript. Finally, there may be specific rituals of collective use and sharing associated with “hashish” that are not with “flowers”.

Tobacco consumption, type of product, use rituals, excess quantity on the market due to the major EU zone of transit… PLOS One has an international audience, not expected to be aware of these elements, and which should therefore be mentioned.

TITLE

I suggest deletion of initial "the" in the title, as in "Exploring Relationships Between".

I also respectfully propose to add language along the lines of: "among 89 Adolescents in Andalusian Cities." at the end.

I have the memory of an original title being perhaps more specific in that regard (but I could not locate it on the editorial manager).

ABSTRACT — Methods

"This study conducted in Andalusia, Spain, surveyed 1,051 adolescents aged 14 to 18 years, of whom 89 reported testing positive for COVID-19."

I continue to believe that the mention of the figure 1,051 is not relevant in the abstract —no more than the totla figure of adolescents in Westedn Andalucia. This is an article about positive results from a study of 89 participants. The abstract must reflect this in a non-misleading way. Given that the rest of the article refers to "our sample" without making clear that sample is 89, not 1,051.

The figure 1,051 is superfluous.

ABSTRACT — Results

"Sharing cannabis during the pandemic and the lack of perception that cannabis use exacerbates COVID-19 are related to cannabis use"

Did the authors intended to mean "is related to cannabis use" (or "related to…") instead of "are"? If so, the meaning changes drastically.

If not, the authors are saying with this sentence:

1) Sharing cannabis during the pandemic … is related to cannabis use — a wholly tautological statement.

2) lack of perception that cannabis use exacerbates COVID-19 … is related to cannabis use — a proper finding.

(1) is superfluous. The fact that (1) is merged with the entirely different (2) is concerning.

ABSTRACT — Conclusions

This section also requires considerable revision and restraint. Particularly, the latter sentence is perhaps superfluous or overstated. This paper does not in any way bring "comprehensive" knoelwedge, just additional elements. I must again note a regrettable tendency towards overstatement throughout.

These are signaling a certain lack of ability to communicate results in an accurate and transparent manner. Abstracts are critical parts of the research. I reiterate my call for a comprehensive rewriting of the abstract and title, keeping in mind modesty and the scope of these results: this is a study of 89 cases in 4 large cities in half of one region of Spain. It is an extremely insightful, interesting, well-conducted, and useful study. But it is not what it is not, and must focus on what it is.

INTRODUCTION §1

This opening is structurally problematic. It starts by talking about the WHO’s opinion on “the pandemic” without having qualified it prior.

In my opinion, this introduction ought to start with the first sentence from the M&M section (i.e., “In March 2020, a lockdown…” up to “… of our adolescents”), followed by the present sentence about WHO and end of pandemic period.

This would be a serious, solid, clear way to begin the article, and would allow to avoid subsequent repetitions (hence reducing the length of the introduction andM&M section).

INTRODUCTION §2

"The global increase in drug seizures in 2020 intensified during the coronavirus disease (COVID-19) pandemic in many countries [5]. In fact, the United Nations' Annual Report for 2020 against drug abuse highlighted that most countries reported an increase in cannabis consumption during the COVID-19 pandemic [6]."

Seizures are only reflective of one thing: police zeal and activity. Stop funding police, seizures go down: does that mean drug use goes down? It is inappropriate to conflate such distinct data in this sentence. Delete the paragraph, for global seizure data serve no analytical purpose. The only use of seizure data may be to address the hashish-contet mentioned elsewhere (but I assume this would not go in §2). Otherwise, this sentence and its discussion of global police activity has no room in a paper about adolescents in four specific urban contexts of Western Andalucía.

Provided that this entire paragraph adds nothing to the topic, and conflates broad data, I do not see any better solution than deleting it.

INTRODUCTION §3

"Cannabis consumption poses a threat to public health, since it stands as the most widely used illicit drug among teenagers globally [7]."

Prevalence alone does not constitute evidence of a “public health threat.” UNODC World Drug Report, the source cited, explicitly distinguishes between prevalence of use and of cannabis use disorders. Cannabis use disorders represent a subset of cannabis consumption. Failing to acknowledge this distinction, which is explicit in the cited source, while conflating widespread use with public health harm, is conceptually incorrect and misleading.

I would like to highlight that this seems to constitute a persistent trend in conflating prevalence, harm, simple use and use disorder, and causality — which apparently remain uncorrected despite repeated and explicit reviewer feedback across multiple rounds of review.

From a broader perspective, this paragraph does not serve the stated aims of the manuscript. The article does not examine cannabis use as a public health risk per se, but rather the association between cannabis use and reporting testing positive for COVID-19. As such, general statements about cannabis as a public health threat unrelated to COVID-19 (or possibly other similar infectious and/or lung-related health conditions) are not analytically relevant in this context.

The paragraph would be better aligned with the study objectives if it focused directly on prior evidence linking cannabis use and COVID-19 outcomes, followed by relevant prevalence data to contextualize the population studied (global → European → Spanish adolescents):

(sentence 1 of the new §) Previous studies, such as that by Griffith et al. (2024), have reported that cannabis use was associated with an increased risk of hospitalization and ICU admission among individuals with COVID-19. (sentence 2 of the new §) The global prevalence of cannabis use in the last year was 5.8% (corresponding to 12 million) among 15- and 16-year-old adolescents [8], with the highest prevalence worldwide observed in European youths aged 15 to 24 [6,9] (sentence 3 of the new §) move on with data about Spain’s consumption patterns among the youth, currently in the next paragraph (4th)

INTRODUCTION §4

Suggested merger with revised §3 as proposed above.

This does not apply to the final part of the paragraph: “the EDADES study (population aged 15 to 64) indicates an increasing trend from the age of 24 onwards”. If this article is about adolescents under 24, there are little reasons to mention this datum.

M&M

This section ought to refer to the specificities of the territory (4 large cities) and the product (resin mixed with tobacco, imports, large market), as mentioned above.

CONCLUSION.

"The findings of this exploratory study give light to possible interventions to reduce the harm of cannabis use in a pandemic situation"

Replace "the harm of cannabis" by "harms associated with cannabis use" you may want to add elements of context. This paper offers limited guidance on reducing the harms of pure flowers vaporised in, say, Hawaii. It serves for tobacco-hashcish mix in adolescents in Europe.

I would not recommend accepting a final version which would retain the mention of "interventions" as it currently stands. This article gives absolutely no insight for practical intervention. Stating so is a large leap. A premature one. It would be concerning if real-world public health interventions were based solely on such cross-sectional studies of small samples of urban adolescents.

Consider something along the lines of: "The findings of this exploratory study provide additional data in relation to the harms associated with the use of cannabis resin among adolescents, in this specific context, in a situation of public health emergency like the COVID-19 pandemic, and may provide insight in the design of public health interventions."

I am conscious of the discomfort this long review process may have caused to the authors, but I am confident that integrating these improvements will make for a paper reduced in its scope, but dramatically increased in its quality — which will benefit both public health and the authors' track record.

I thank you and am confident the authors' revised version will be good for publication.

Reviewer #3: I am happy having the opportunity to read and review the manuscript assigned to me, titled “Exploring the Relationships Between Reporting Testing Positive for COVID-19 and Cannabis Use: an Exploratory Study” submitted to Plos One.

I have carefully reviewed this manuscript and below is my decision.

The paper is well-suited for publication in the Plos One. All comments are well-improved the quality of paper. Good luck!

7. PLOS authors have the option to publish the peer review history of their article (what does this mean?). If published, this will include your full peer review and any attached files.

Reviewer #1: **Yes:**Cláudio Calixto Carlos da Silva

Reviewer #2: No

Reviewer #3: No

---

## [Author Response · Author response to Decision Letter 3]

16 Mar 2026

Please, find all of this information in the uploaded document titled "response to reviewers" for an easier review.

Reviewer #1:

I would like to thank the authors for the substantial improvements made throughout the revision process. The manuscript is now clearer, more consistent, and the main concerns raised in previous rounds have been adequately addressed. The adjustments to causal language, clarification of the introduction, and revisions throughout the text have strengthened the overall quality of the work.

At this point, I believe the manuscript is suitable for publication pending only minor formatting adjustments. Several tables still present formatting inconsistencies, including:

Table 1 appearing split across two pages;

Placeholder text such as “insertheretable1”, “insertheretable2”, and “ insertheretable3” remaining in the manuscript.

I kindly recommend a final formatting check to ensure that all tables appear as single, complete blocks and that placeholders are removed. These adjustments are minor and do not affect the scientific content of the manuscript.

Response from authors:

We thank reviewer 1 for their comments. We have removed the 'insert Table 1 here' in all of the table’s placeholder, and Table 1 now appears on a single page.

Reviewer #2:

I thank the authors for their work and for their involvement in the review process. I also take the opportunity to wish them all an happy new year.

I note the authors have "carefully reviewed the entire manuscript to ensure consistent use of language that accurately reflects the cross-sectional nature of the study and avoids any implication of causality." While this should not have required two rounds of peer-review, it is indeed finally appropriate in the body of the manuscript; it is, however not, in the Abstract. My review focuses therefore on the Abstract, as well as the introduction.

An outstanding element that deserve address is the apparent deterritorialisation of the results, which does not seem justified. This study was conducted in four specific urban contexts of Western Andalucía, and among adolescents. None of these elements are present in the title (geographical location [Andalucia], type of environment [urban], population studied [scholarised adolescents]). It is critical that these elements appear in the title, and clearly articulated in the abstract.

The importance of accurately representing the specificities of the localised regional context within which the study was conducted (urban context in Western Andalucía) is key.

I also stress that the territory in question (four specific large cities in Western Andalucía) is located at the najor entry point of cannabis resin for illicit markets, fromNorthern Africa to Europe. This absolutely unique situation is nowhere mentioned. Other geographical, socio-cultural, and political contextual elements may have been welcome.

Furthermore, I note that imported cannabis drugs from that Northern African region often constitute “hashish” (cannabis resin) and not “flowers” (dry herbal tops) which are the subject of most published research. Besides possible differences in effects on individual and public health, the consumption of hashish is, I believe, directly associated with that of tobacco. I may have missed it, but this critical element was not mentioned in the manuscript. Finally, there may be specific rituals of collective use and sharing associated with “hashish” that are not with “flowers”.

Tobacco consumption, type of product, use rituals, excess quantity on the market due to the major EU zone of transit… PLOS One has an international audience, not expected to be aware of these elements, and which should therefore be mentioned.

TITLE

I suggest deletion of initial "the" in the title, as in "Exploring Relationships Between".

I also respectfully propose to add language along the lines of: "among 89 Adolescents in Andalusian Cities." at the end.

I have the memory of an original title being perhaps more specific in that regard (but I could not locate it on the editorial manager).

ABSTRACT — Methods

"This study conducted in Andalusia, Spain, surveyed 1,051 adolescents aged 14 to 18 years, of whom 89 reported testing positive for COVID-19."

I continue to believe that the mention of the figure 1,051 is not relevant in the abstract —no more than the totla figure of adolescents in Westedn Andalucia. This is an article about positive results from a study of 89 participants. The abstract must reflect this in a non-misleading way. Given that the rest of the article refers to "our sample" without making clear that sample is 89, not 1,051.

The figure 1,051 is superfluous.

ABSTRACT — Results

"Sharing cannabis during the pandemic and the lack of perception that cannabis use exacerbates COVID-19 are related to cannabis use"

Did the authors intended to mean "is related to cannabis use" (or "related to…") instead of "are"? If so, the meaning changes drastically.

If not, the authors are saying with this sentence:

1) Sharing cannabis during the pandemic … is related to cannabis use — a wholly tautological statement.

2) lack of perception that cannabis use exacerbates COVID-19 … is related to cannabis use — a proper finding.

(1) is superfluous. The fact that (1) is merged with the entirely different (2) is concerning.

ABSTRACT — Conclusions

This section also requires considerable revision and restraint. Particularly, the latter sentence is perhaps superfluous or overstated. This paper does not in any way bring "comprehensive" knoelwedge, just additional elements. I must again note a regrettable tendency towards overstatement throughout.

These are signaling a certain lack of ability to communicate results in an accurate and transparent manner. Abstracts are critical parts of the research. I reiterate my call for a comprehensive rewriting of the abstract and title, keeping in mind modesty and the scope of these results: this is a study of 89 cases in 4 large cities in half of one region of Spain. It is an extremely insightful, interesting, well-conducted, and useful study. But it is not what it is not, and must focus on what it is.

INTRODUCTION §1

This opening is structurally problematic. It starts by talking about the WHO’s opinion on “the pandemic” without having qualified it prior.

In my opinion, this introduction ought to start with the first sentence from the M&M section (i.e., “In March 2020, a lockdown…” up to “… of our adolescents”), followed by the present sentence about WHO and end of pandemic period.

This would be a serious, solid, clear way to begin the article, and would allow to avoid subsequent repetitions (hence reducing the length of the introduction andM&M section).

INTRODUCTION §2

"The global increase in drug seizures in 2020 intensified during the coronavirus disease (COVID-19) pandemic in many countries [5]. In fact, the United Nations' Annual Report for 2020 against drug abuse highlighted that most countries reported an increase in cannabis consumption during the COVID-19 pandemic [6]."

Seizures are only reflective of one thing: police zeal and activity. Stop funding police, seizures go down: does that mean drug use goes down? It is inappropriate to conflate such distinct data in this sentence. Delete the paragraph, for global seizure data serve no analytical purpose. The only use of seizure data may be to address the hashish-contet mentioned elsewhere (but I assume this would not go in §2). Otherwise, this sentence and its discussion of global police activity has no room in a paper about adolescents in four specific urban contexts of Western Andalucía.

Provided that this entire paragraph adds nothing to the topic, and conflates broad data, I do not see any better solution than deleting it.

INTRODUCTION §3

"Cannabis consumption poses a threat to public health, since it stands as the most widely used illicit drug among teenagers globally [7]."

Prevalence alone does not constitute evidence of a “public health threat.” UNODC World Drug Report, the source cited, explicitly distinguishes between prevalence of use and of cannabis use disorders. Cannabis use disorders represent a subset of cannabis consumption. Failing to acknowledge this distinction, which is explicit in the cited source, while conflating widespread use with public health harm, is conceptually incorrect and misleading.

I would like to highlight that this seems to constitute a persistent trend in conflating prevalence, harm, simple use and use disorder, and causality — which apparently remain uncorrected despite repeated and explicit reviewer feedback across multiple rounds of review.

From a broader perspective, this paragraph does not serve the stated aims of the manuscript. The article does not examine cannabis use as a public health risk per se, but rather the association between cannabis use and reporting testing positive for COVID-19. As such, general statements about cannabis as a public health threat unrelated to COVID-19 (or possibly other similar infectious and/or lung-related health conditions) are not analytically relevant in this context.

The paragraph would be better aligned with the study objectives if it focused directly on prior evidence linking cannabis use and COVID-19 outcomes, followed by relevant prevalence data to contextualize the population studied (global → European → Spanish adolescents):

(sentence 1 of the new §) Previous studies, such as that by Griffith et al. (2024), have reported that cannabis use was associated with an increased risk of hospitalization and ICU admission among individuals with COVID-19. (sentence 2 of the new §) The global prevalence of cannabis use in the last year was 5.8% (corresponding to 12 million) among 15- and 16-year-old adolescents [8], with the highest prevalence worldwide observed in European youths aged 15 to 24 [6,9] (sentence 3 of the new §) move on with data about Spain’s consumption patterns among the youth, currently in the next paragraph (4th)

INTRODUCTION §4

Suggested merger with revised §3 as proposed above.

This does not apply to the final part of the paragraph: “the EDADES study (population aged 15 to 64) indicates an increasing trend from the age of 24 onwards”. If this article is about adolescents under 24, there are little reasons to mention this datum.

M&M

This section ought to refer to the specificities of the territory (4 large cities) and the product (resin mixed with tobacco, imports, large market), as mentioned above.

CONCLUSION.

"The findings of this exploratory study give light to possible interventions to reduce the harm of cannabis use in a pandemic situation"

Replace "the harm of cannabis" by "harms associated with cannabis use" you may want to add elements of context. This paper offers limited guidance on reducing the harms of pure flowers vaporised in, say, Hawaii. It serves for tobacco-hashcish mix in adolescents in Europe.

I would not recommend accepting a final version which would retain the mention of "interventions" as it currently stands. This article gives absolutely no insight for practical intervention. Stating so is a large leap. A premature one. It would be concerning if real-world public health interventions were based solely on such cross-sectional studies of small samples of urban adolescents.

Consider something along the lines of: "The findings of this exploratory study provide additional data in relation to the harms associated with the use of cannabis resin among adolescents, in this specific context, in a situation of public health emergency like the COVID-19 pandemic, and may provide insight in the design of public health interventions."

I am conscious of the discomfort this long review process may have caused to the authors, but I am confident that integrating these improvements will make for a paper reduced in its scope, but dramatically increased in its quality — which will benefit both public health and the authors' track record.

I thank you and am confident the authors' revised version will be good for publication.

Response from authors:

We sincerely thank the reviewer for the careful, detailed, and constructive feedback. We appreciate the emphasis on contextual precision, conceptual clarity, and restraint in interpretation. Below we respond point by point and indicate how the manuscript has been revised accordingly.

1. Title

We have:

• Removed the initial “The” as suggested.

• Revised the title to better reflect the specific population and scope.

• Included clarification regarding the study sample.

The new title reads: Association Between Reporting Testing Positive for COVID-19 and Cannabis Use Among Adolescents in Andalusia: an Exploratory Study

2. Abstract

Methods (Sample Size)

We appreciate the concern regarding potential ambiguity.

To improve clarity and avoid any perception of misleading framing:

• In the method section of the abstract states that the analyses focus on the 89 adolescents who reported testing positive.

• The total surveyed population (1,051) is no longer foregrounded in a way that could obscure the analytical sample.

• The wording has been revised to ensure that “our sample” consistently refers to the 89 participants analyzed.

Results (Grammar and Tautology)

Thank you for highlighting this issue. The sentence has been rewritten to avoid ambiguity and tautology. We have removed the phrasing that could be interpreted as circular.

The revised sentence now states:

Individuals testing positive for COVID-19 are almost 2.89 times more likely to have used cannabis in the last month. Sharing cannabis among adolescents during the pandemic may increase the risk of COVID-19 transmission. Additionally, lack of perceived risk that cannabis use could exacerbate COVID-19 was associated with cannabis use during the pandemic.

Conclusions (Overstatement)

We agree that the previous wording overstated the contribution.

The conclusion has been substantially rewritten to reflect:

• The exploratory and cross-sectional nature of the study

• The limited sample size

• The contextual specificity

Any reference to “comprehensive knowledge” has been removed. The conclusions are now framed as providing additional contextualized evidence rather than broad claims.

It is reflected as follows:

Informing whether adolescents who use cannabis may be more likely to test positive for COVID-19 contributes to a better understanding of substance use patterns during a public health emergency. Given the exploratory and cross-sectional nature of this study, as well as its limited and context-specific sample, the findings should be interpreted with caution. The results offer additional contextualized evidence that may help inform future research on the relationship between COVID-19-related factors and cannabis use among adolescents in Western Andalusia.

3. Introduction — Structure (§1)

We agree with the structural suggestion.

The Introduction has been reorganized so that:

• The contextual description of lockdown in March 2020 now appears at the beginning.

• The WHO declaration is introduced after situating the timeline.

• Redundancies between Introduction and Methods have been eliminated.

This restructuring has improved clarity and reduced repetition.

Introduction — Drug Seizure Data (§2)

We accept the reviewer’s concern regarding the interpretative limitations of seizure data.

The paragraph referencing global seizure trends has been removed, as it did not directly serve the analytical aims of the manuscript.

Introduction — Public Health Threat and Conceptual Distinctions (§3)

We appreciate this important conceptual clarification.

The paragraph has been revised to:

• Avoid conflating prevalence with harm or use disorder.

• Remove generalized statements framing cannabis use as inherently a “public health threat.”

• Focus directly on prior evidence linking cannabis use with COVID-19 outcomes.

• Present prevalence data in a structured progression, as suggested.

We agree that this revision better aligns the Introduction with the study objective.

EDADES Data Reference

We have removed the reference to trends in populations aged 24+ years, as it was not directly relevant to an adolescent sample.

Contex

---

## [Decision Letter · Decision Letter 3]

30 Mar 2026

Association Between Cannabis Use and Result of COVID-19 Testing Among Scholarised Adolescents in Andalusia: a Cross-Sectional Study

PONE-D-24-49229R3

Dear Dr. Baena-Jiménez,

We’re pleased to inform you that your manuscript has been judged scientifically suitable for publication and will be formally accepted for publication once it meets all outstanding technical requirements.

Kind regards,

Vincenzo De Luca

Academic Editor

PLOS One

Additional Editor Comments (optional):

Reviewers' comments:

Reviewer's Responses to Questions

**Comments to the Author**

1. If the authors have adequately addressed your comments raised in a previous round of review and you feel that this manuscript is now acceptable for publication, you may indicate that here to bypass the “Comments to the Author” section, enter your conflict of interest statement in the “Confidential to Editor” section, and submit your "Accept" recommendation.

Reviewer #1: All comments have been addressed

Reviewer #2: All comments have been addressed

Reviewer #3: All comments have been addressed

2. Is the manuscript technically sound, and do the data support the conclusions?

Reviewer #1: Yes

Reviewer #2: Yes

Reviewer #3: Yes

3. Has the statistical analysis been performed appropriately and rigorously? 

Reviewer #1: Yes

Reviewer #2: Yes

Reviewer #3: Yes

4. Have the authors made all data underlying the findings in their manuscript fully available?

Reviewer #1: Yes

Reviewer #2: Yes

Reviewer #3: Yes

5. Is the manuscript presented in an intelligible fashion and written in standard English?

Reviewer #1: Yes

Reviewer #2: Yes

Reviewer #3: Yes

6. Review Comments to the Author

Reviewer #1: I would like to thank the authors for their careful revisions and for adequately addressing all comments raised in the previous review round. The manuscript has improved substantially in clarity, consistency, and overall presentation.

The previously noted formatting issues have been corrected, including the proper placement of tables and removal of placeholder text. These adjustments resolve the remaining concerns.

At this stage, I have no further comments. I consider the manuscript scientifically sound, well presented, and fully suitable for publication in its current form.

Reviewer #2: Thank you.

Reviewer #3: I am happy having the opportunity to read and review the manuscript assigned to me, titled “Association Between Cannabis Use and Result of COVID-19 Testing Among Scholarised Adolescents in Andalusia: a Cross-Sectional Study” submitted to Plos One.

I have carefully reviewed this manuscript and below is my decision.

The paper is well-suited for publication in the Plos One. All comments are well-improved the quality of paper. Good luck!

7. PLOS authors have the option to publish the peer review history of their article (what does this mean?). If published, this will include your full peer review and any attached files.

Reviewer #1: **Yes:**Cláudio Calixto Carlos da Silca

Reviewer #2: No

Reviewer #3: No

---

## [Editor Report · Acceptance letter]

PONE-D-24-49229R3

PLOS One

Dear Dr. Baena-Jiménez,

I'm pleased to inform you that your manuscript has been deemed suitable for publication in PLOS One. Congratulations! Your manuscript is now being handed over to our production team.

Kind regards,

on behalf of

Dr. Vincenzo De Luca

Academic Editor

PLOS One